# COVID-19 could accelerate the decline in recreational hunting: A natural experiment from Northern Italy

**Jacopo Cerri[1], Carmela Musto[2], Marco Ferretti[3], Mauro Delogu[2], Sandro Bertolino[4], Adriano Martinoli[5], Francesco Bisi[5]\*, Damiano Giovanni Preatoni[5], Clara Tattoni[5], Marco Apollonio[1]**

1 Department of Veterinary Medicine, University of Sassari, Sassari, Italy, 2 Department of Veterinary Medical Sciences, University of Bologna, Bologna, Italy, 3 Regione Toscana, Firenze, Italy, 4 Dipartimento di Scienze della Vita e Biologia dei Sistemi, Università degli Studi di Torino, Torino, Italy, 5 Department of Theoretical and Applied Sciences, Università degli Studi dell'Insubria, Varese, Italy

\* francesco.bisi@uninsubria.it

**Data Availability Statement:** A reproducible dataset and software code are available at https://osf.io/j25cr/.

**Funding:** The co-author Carmela Musto was partially supported by a research grant funded by

## Abstract

Although many studies highlighted the potential of COVID-19 to reshape existing models of wildlife management, empirical research on this topic has been scarce, particularly in Europe. We investigated the potential of COVID-19 pandemic to accelerate the ongoing decline in an aging population of recreational hunters in Italy. Namely, we modelled spatio-temporal trends between 2011 and 2021 in the number of recreational hunters in 50 Italian provinces with a varying incidence of COVID-19, and temporally delayed waves of infection. Compared to projections from 2011–2019 data, we detected a lower number of hunters who enrolled for the hunting season, both in 2020 (14 provinces) and in 2021 (15 provinces). The provinces with the highest incidence of COVID-19 in the Lombardy and Emilia-Romagna regions were also those experiencing the most marked decrease in hunting participation. Our findings revealed that a wildlife management system based on recreational hunting can be rapidly destabilized by epidemics and their associated public health measures, particularly when the average age of hunters is high, like in Italy. Considered the high incidence attained by COVID-19 in many European countries, where hunters are pivotal for the management of large ungulates and where they were already declining before the pandemic, our findings call for further large-scale research about the impact of COVID-19 on hunting participation.

## Introduction

Between early 2020 and early 2023, Sars-CoV-2 infected over 655 million people globally, leading to more than 6 million official deaths [1] and a much higher excess mortality [2]. Millions of people experienced long-term physical and mental harm and COVID-19 caused temporary global reductions in human mobility [3].

In conservation science, the COVID-19 pandemic was studied from multiple viewpoints [4, 5]. Some studies explored how changes in human mobility affected the ecology (e.g., diet,

the Vienna Science and Technology Fund (WWTF) [10.47379/ESR20009], the funders had no role in study design, data collection and analysis, decision to publish, or preparation of the manuscript.

**Competing interests:** The authors have declared that no competing interests exist.

[6, 7]), demographics [8] and behaviour of wildlife (e.g., movement, [9–11]). Other research quantified changes in human-nature relationships [12] and explored the potential effects of the pandemic on existing management models for wildlife and ecosystems [13].

Some other studies suggested that the sustained impacts of COVID-19 pandemic on outdoor activities [14–16], could potentially result in long-lasting changes in patterns of outdoor recreation [17], including recreational hunting. This consequence is plausible from both demographic and behavioural viewpoints. In facts, COVID-19 resulted into a significant increase in the death rate or health complications among people above 65 years of age [18, 19]. Consequently, the pandemic may have prompted many hunters to prematurely end their hunting career due to health problems, caregiving responsibilities for older relatives, or even higher mortality among older hunters [20, 21]. Moreover, many countries repeatedly enforced restrictions to human mobility, whenever they experienced peaks in SARS-CoV2 incidence and deaths from COVID-19, and some even halted the hunting season (e.g., Portugal [22]). This could have increased the uncertainty of hunters about the upcoming hunting seasons in 2020 and 2021, potentially leading them to not renew their licenses.

While both mechanisms are possible, and despite some research has been conducted about recreational angling [23], empirical research on the impact of COVID-19 on hunting licenses has been limited to two studies, carried out in North America. Namely, Chizinski et al. [24] found a decrease of approximately 90% in permits for non-resident turkey hunters across the US in 2020 compared to the previous three years. Conversely, Danks et al. [25] reported an increase in participation in turkey hunting across the US in 2020, aligning with grey literature that indicated an overall increase in hunting licenses across the US in 2020 and 2021 [26].

Given the critical role that hunters play in the management of wildlife in Europe, the lack of studies in this continent is cause for concern. The monitoring [27, 28], control [29] and epidemiological surveillance [30] of wildlife populations are among the many tasks that hunters are responsible for in many nations.

In this study we investigated the potential of COVID-19 pandemic to accelerate the ongoing decline in recreational hunting, using Italy as a natural experiment. In this country ungulate hunters are responsible of extensive monitoring to gain data necessary to produce management plans and COVID 19 prevented these activities during 2020 and 2021. Northern Italy was the first epicentre of 2020 COVID-19 outbreak in Europe [31] and Italian regions differed in the temporal progression of the pandemic, as well as in the implementation of restrictions on crowding and human mobility [32]. This allowed us to: *i*) model spatiotemporal trends in the number of recreational hunters (2011–2019) across 50 Italian provinces with varying incidence of SARS-CoV2, and temporally delayed waves of infection, *ii*) detect discrepancies between the observed and predicted number of hunters in 2020 and 2021 and *iii*) correlate these discrepancies with spatial differences in the evolution of the COVID-19 pandemic.

## Materials and methods

The study area encompassed 50 provinces across the regions of Emilia-Romagna, Friuli Venezia-Giulia, Lombardy, Piedmont, Veneto, and Tuscany in Central and Northern Italy (Fig 1). Overall, these regions housed 28.4 million residents in 2020, accounting for 47.7% of the Italian population.

In Italy, professional hunting does not exist, and hunting is a recreational activity. Wildlife is considered a public property and falls under the ownership of the state. During each hunting season, which typically spans from September to January, hunters are required to pay a tax for hunting license to the Italian government and the region where they reside and then a tax to

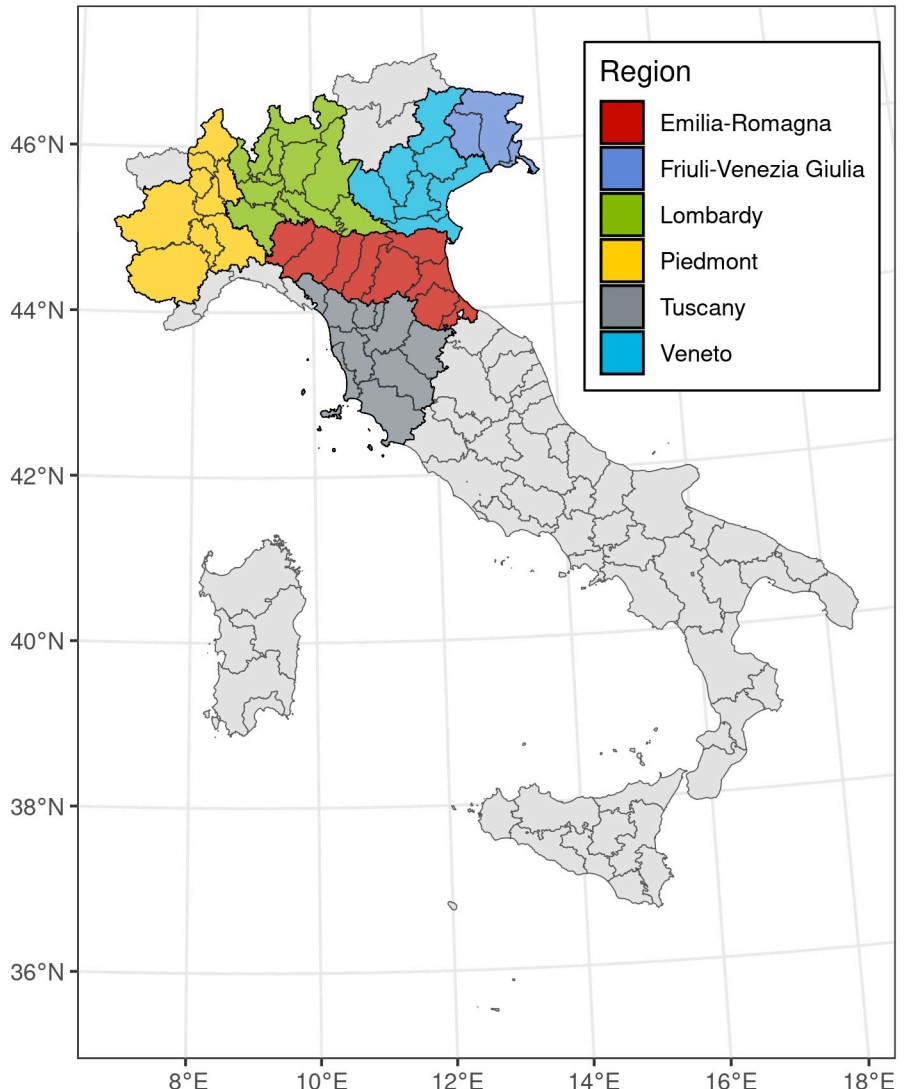

**Fig 1. Map of the study area, representing the 50 provinces, divided between the 6 different regions.**
Administrative boundaries were downloaded from the Italian National Institute of Statistics and are subjected to a
CCBY 4.0 license (https://www.istat.it/note-legali/).

the hunting area/s (i.e. Ambiti Territoriali di Caccia and Comprensori Alpini) where they
hunt. While recreational hunting has focused in the past on small game and migratory passer-
ines [33], there has been a consistent rise in the abundance and exploitation of wild ungulates
since the early 2000s [34]. The annual hunting bags for these species are nowadays consider-
able exceeding 500.000 heads [35].

Over the past four decades, Italy has experienced a considerable long-term decline in rec-
reational hunting. In 1980 there were 1,701,853 hunters, whereas in 2017 the Ministry of the
Interior estimated a total of 678,970 hunting licenses. The overall number of hunters in the
study area fell by 37.8% in Emilia-Romagna, 29.1% in Friuli Venezia-Giulia, 33.2% in Lom-
bardy, 36.9% in Piedmont, 44.6% in Tuscany, and 29.2% in the Veneto region, between 2004
and 2019. In urbanised areas, this loss was more noticeable [36]. In addition, recreational

hunters had been aging. In 2010 the percentage of hunters above 60 years of age was approximately 52% in Tuscany and Emilia-Romagna region [37, 38], and 43% in the Lombardy region [39].

In 2020, the evolution of COVID-19 varied across the five regions in the study area. The initial wave, occurring in spring 2020, predominantly affected Lombardy and parts of the Emilia-Romagna and Piedmont regions, with no significant excess deaths observed in Friuli Venezia-Giulia, Veneto, and Tuscany regions [40–42]. The spread of the virus was mitigated by the nationwide implementation of non-pharmaceutical measures [43], from March to May, possibly influenced by the progressive increase in air temperatures [44]. However, a second wave of COVID-19 emerged in autumn 2020, affecting the entire study area, and excess deaths became widespread throughout 2021 [45]. Despite the imposition of various restrictions on human mobility throughout 2020 and 2021, the hunting season was regularly opened in September during both years.

To assess the impact of COVID-19 on hunting participation, we gathered data on the number of hunting licenses issued by each province from 2011 to 2019. Hunting licenses were employed as a proxy for hunting participation, recognizing that some hunters may have renewed their licenses without actively participating in the season. In Italy the hunting season starts in September; consequently, we quantified the number of licenses for the 2020/2021 season (hereinafter "2020") and the 2021/2022 season (hereinafter "2021").

We used Bayesian Generalized Linear Models to capture spatiotemporal variations in the number of hunters in the study area. Specifically, we used a negative binomial distribution to model the annual number of hunters in each province. Model selection was based on leave-one-out cross validation, and diagnostic assessment followed the approach outlined by Zuur et al. [46]. Overall, we tried multiple spatio-temporal structures to account for different potential spatiotemporal patterns, as suggested by Blangiardo & Cameletti [47]. Model selection retained a best candidate model where the number of hunters was modelled according to a non-linear random walk term, with a random intercept assigned to each region, to account for differences in the total number of residents across provinces. We also accounted for temporal trends between 2011 and 2019 through a linear term.

Finally, we compared the total number of hunters in each province, during 2020 and 2021, against predicted values from the posterior distribution of our model, trained on 2011–2019 data. We considered anomalies those values that exceeded the boundaries of the 95% credibility interval. Additionally, we compared the number of hunters in 2020 and 2021 against the median value of the posterior distribution, which in Bayesian models represent the most likely predicted value [48]. Regarding the Veneto region, we could only test for anomalies for the 2020/2021 season, because the number of hunting licenses for 2021 was not available at the time of the study.

Statistical models were implemented using INLA [49] and the "inlabru" package in R [50]. A reproducible dataset and software code are available at https://osf.io/j25cr/

## Results

Our findings indicate that the number of hunters who enrolled for the hunting season was lower than expected based on the 2011–2019 trend in 14 Italian provinces during 2020 and in 15 provinces in 2021. Anomalous decreases were observed specifically in provinces within the Lombardy, Emilia-Romagna, and Tuscany regions (Fig 2).

Lombardy exhibited the most pronounced decrease in both 2020 and 2021. By comparing the observed number of hunters in Lombardy with the median value of the posterior distribution, it was found that there were 3,904 (2,936–4,911, compared to the upper and lower bound

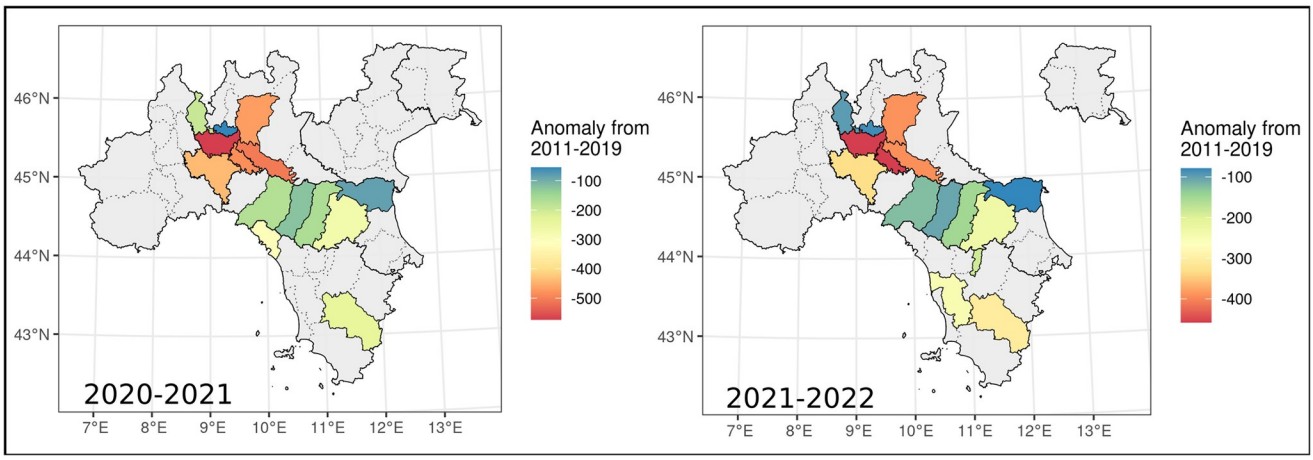

**Fig 2. Difference between the observed number of hunting licenses and predictions from 2011–2019 data, during the 2020–2021 season (left) and the 2021–2022 season (right).** Anomalies are expressed as the difference between the median value of the posterior distribution, which in Bayesian models represent the most probable value, and observed values. Provinces in grey did not have any anomalous variation. Administrative boundaries were downloaded from the Italian National Institute of Statistics and are subjected to a CCBY 4.0 license (https://www.istat.it/note-legali/).

of 95% Credibility Interval) fewer hunters in 2021 and 2,720 (1,790–3,688) fewer in 2020. There were 808 fewer hunters in the Emilia-Romagna region in 2020 (255–1,382) and 682 in 2021 (138–1,249). There were 538 fewer hunters in the Tuscany region in 2020 (259–828) and 741 fewer in 2021 (213–1,291), however only two out of ten regions in 2020 and three provinces in 2021 were affected by these anomalies. A complete overview of the decrease in hunters for each province is shown in Table 1.

The province of Milan exhibited the most significant decrease overall (396–756 fewer hunters in 2020 and 284–638 in 2021), followed by Cremona (399–619 fewer hunters in 2020 and 286–503 in 2021), Lodi (402–570 fewer hunters in 2020 and 377–541 in 2021) and Bergamo (175–774 fewer hunters in 2020 and 101–691 in 2021).

According to the Italian National Institute of Statistics, the Lombardy and Emilia-Romagna regions exhibited the highest excess mortality in 2020, compared to the 2015–2019 average [45] and provinces with negative anomalies in hunting licenses concentrated in these areas (Table 1, Fig 3).

In 2020 the decrease in licences correlated positively with excess mortality (Spearman's rho = 0.52, panel a of Fig 4); this relationship was negative in 2021 (Spearman's rho = -0.62, panel a of Fig 4), and the decrease in licences during 2021 correlated positively with 2020 excess mortality (Spearman's rho = 0.43, panel c of Fig 4).

## Discussion

Although many studies highlighted the potential of the COVID-19 pandemic to reshape socio-ecological systems [51], the amount of empirical research quantifying its implications on wildlife management models has been relatively limited. Through an examination of regional differences in COVID-19 progression in Northern Italy, we found that areas of the country experiencing the most immediate and pronounced impacts from COVID-19 also exhibited an anomalous decline in the number of hunting licenses in 2020 and 2021. We believe our study holds important implications to understand how wildlife management in Italy and Europe can be impacted by large scale epidemic events, and it calls for further large-scale collaborative research on trends in recreational hunting across Europe.

**Table 1. Overview of predicted and observed number of hunters in the 2020/2021 and 2021/2022 hunting season.** Only provinces with an anomalous decrease are included in the table. Excess mortality was obtained from the Italian National Institute of Statistics [45] and it was represented as the percentage increase from the mean mortality in 2015–2019.

**Hunting season: 2020/2021**

| Province | Region | Excess mortality compared to 2015–2019 (%) | Number licences (observed) | Number licences (predicted, most likely) | Number licences (predicted, min-max) | Anomalous decrease (compared to the median) | | Anomalous decrease (min-max) | |
|---|---|---|---|---|---|---|---|---|---|
| | | | | | | Number | Percentage | Number | Percentage |
| Milan | Lombardy | + 32% | 5067 | 5640 | 5463–5823 | 573 | -10.20% | 396–756 | -7.20% / -13.00% |
| Cremona | Lombardy | + 52% | 2879 | 3386 | 3278–3498 | 507 | - 5.00% | 399–619 | -12.20% / -17.70% |
| Lodi | Lombardy | + 47% | 1176 | 1660 | 1578–1746 | 484 | -29.20% | 402–570 | -25.50% / -32.60% |
| Bergamo | Lombardy | + 60% | 9061 | 9530 | 9236–9835 | 469 | -4.90% | 175–774 | -1.90% / -7.90% |
| Pavia | Lombardy | + 32% | 3726 | 4166 | 4034–4303 | 440 | -10.60% | 308–577 | -7.60% / -13.40% |
| Massa Carrara | Tuscany | + 16% | 1490 | 1792 | 1732–1854 | 302 | -16.90% | 242–364 | -14.00% / -19.60% |
| Bologna | Emilia-Romagna | + 13% | 4882 | 5152 | 4990–5320 | 270 | -5.20% | 108–438 | -2.20% / -8.20% |
| Siena | Tuscany | +1% | 6824 | 7060 | 6841–7288 | 236 | -3.30% | 17–464 | -0.20% / -6.40% |
| Varese | Lombardy | + 27% | 2460 | 2653 | 2567–2742 | 193 | -7.30% | 107–282 | -4.20% / -10.30% |
| Parma | Emilia-Romagna | + 31% | 3274 | 3444 | 3330–3563 | 170 | -4.90% | 56–289 | -1.70% / -8.10% |
| Modena | Emilia-Romagna | + 16% | 3496 | 3659 | 3543–3780 | 163 | -4.50% | 47–284 | -1.30% / -7.50% |
| Reggio Emilia | Emilia-Romagna | + 15% | 2924 | 3045 | 2947–3146 | 121 | -4.00% | 23–222 | -0.80% / -7.10% |
| Ferrara | Emilia-Romagna | + 8% | 1811 | 1895 | 1832–1960 | 84 | -4.40% | 21–149 | -1.10% / -7.60% |
| Monza and Brianza | Lombardy | + 33% | 1519 | 1573 | 1520–1629 | 54 | -3.40% | 2–110 | -0.10% / -6.80% |

**Hunting season: 2021/2022**

| Province | Region | Excess mortality compared to 2015–2019 (%) | Number licences (observed) | Number licences (predicted, most likely) | Number licences (predicted, min-max) | Anomalous decrease (compared to the median) | | Anomalous decrease (min-max) | |
|---|---|---|---|---|---|---|---|---|---|
| | | | | | | Number | Percentage | Number | Percentage |
| Milan | Lombardy | + 8% | 4991 | 5449 | 5275–5629 | 458 | -8.41% | 284–638 | -5.38% / -11.33% |
| Lodi | Lombardy | + 5% | 1147 | 1604 | 1524–1688 | 457 | -28.49% | 377–541 | -24.74% / -32.05% |
| Cremona | Lombardy | + 2% | 2879 | 3271 | 3165–3382 | 392 | -11.98% | 286–503 | -9.04% / -14.87% |
| Bergamo | Lombardy | + 2% | 8817 | 9207 | 8918–9508 | 390 | -4.24% | 101–691 | -1.13% / -7.27% |
| Pavia | Lombardy | + 5% | 3694 | 4025 | 3895–4160 | 331 | -8.22% | 201–466 | -5.16% / -11.20% |
| Siena | Tuscany | + 7% | 6514 | 6821 | 6605–7046 | 307 | -4.50% | 91–532 | -1.38% / -7.55% |
| Pisa | Tuscany | + 7% | 7160 | 7408 | 7174–7651 | 248 | -3.35% | 14–491 | -0.20% / -6.42% |
| Bologna | Emilia-Romagna | + 9% | 4749 | 4977 | 4818–5143 | 228 | -4.58% | 69–394 | -1.43% / -7.66% |
| Prato | Tuscany | + 21% | 2186 | 2372 | 2294–2454 | 186 | -7.84% | 108–268 | -4.71% / -10.92% |
| Modena | Emilia-Romagna | + 9% | 3387 | 3536 | 3421–3655 | 149 | -4.21% | 34–268 | -0.99% / -7.33% |
| Parma | Emilia-Romagna | + 5% | 3205 | 3328 | 3215–3445 | 123 | -3.70% | 10–240 | -0.31% / -6.97% |
| Reggio Emilia | Emilia-Romagna | + 7% | 2838 | 2942 | 2846–3042 | 104 | -3.54% | 8–204 | -0.28% / -6.71% |
| Varese | Lombardy | + 13% | 2467 | 2563 | 2479–2651 | 96 | -3.75% | 12–184 | -0.48% / -6.94% |
| Monza and Brianza | Lombardy | + 12% | 1437 | 1520 | 1468–1574 | 84 | -5.53% | 31–137 | -0.02% / -8.70% |
| Ferrara | Emilia-Romagna | + 10% | 1752 | 1830 | 1769–1895 | 78 | -4.26% | 17–143 | -0.96% / -7.55% |

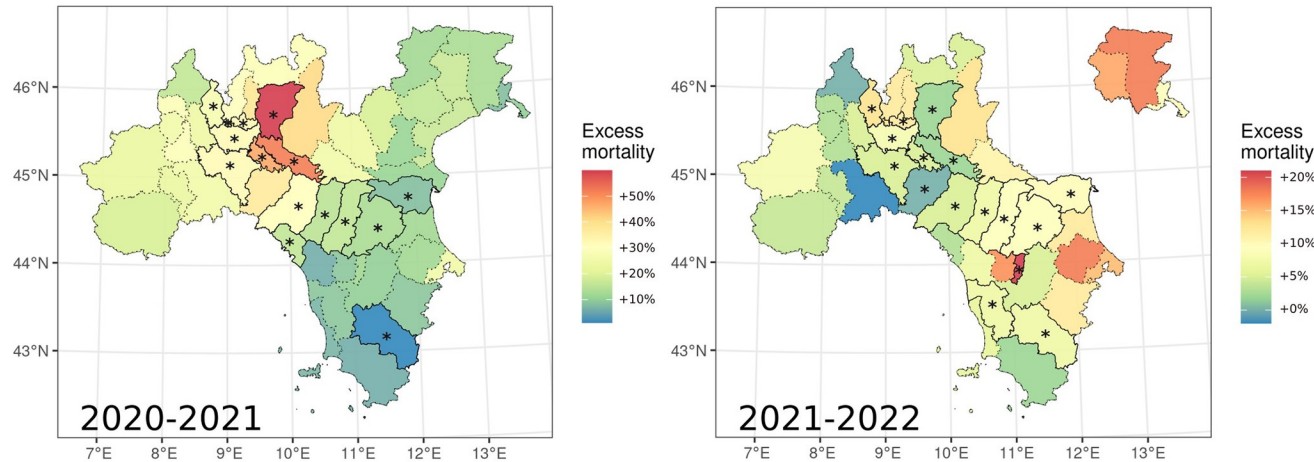

**Fig 3. Overlap between provinces with an anomalous decrease in hunting licences and excess mortality, for 2020–2021 and 2021–2022.** Anomalies are expressed as the difference between the median value of the posterior distribution, which in Bayesian models represent the most probable value, and observed values. Provinces with anomalous values were highlighted and marked with an asterisk. Excess mortality is represented as the percentage increase from the mean mortality in 2015–2019: for the 2020–2021 hunting season 2020 excess mortality was used, while 2021 excess mortality was used for the 2021/2022 season. Administrative boundaries were downloaded from the Italian National Institute of Statistics and are subjected to a CCBY 4.0 license (https://www.istat.it/note-legali/).

Our findings revealed that a wildlife management system where recreational hunters are aging, can be rapidly destabilized by epidemics and the associated public health measures. Lombardy and Emilia-Romagna regions exhibited high excess mortality in 2020 compared to the 2015–2019 average [45]. Interestingly, most provinces with negative anomalies in hunting licenses are concentrated in these regions (Fig 3), suggesting a potential decrease in hunting activity related to COVID-19 diffusion, particularly during 2020. We believe this pattern to have resulted from at least four different mechanisms.

On the one hand, since COVID-19 predominantly affected people above 60 years of age [52], older hunters could have died from COVID-19, or from consequence of the lack of health care deriving from it (e.g., from missed diagnoses [53]). The Lombardy and Emilia-Romagna regions were characterized by an impressive number of excess deaths and recoveries in

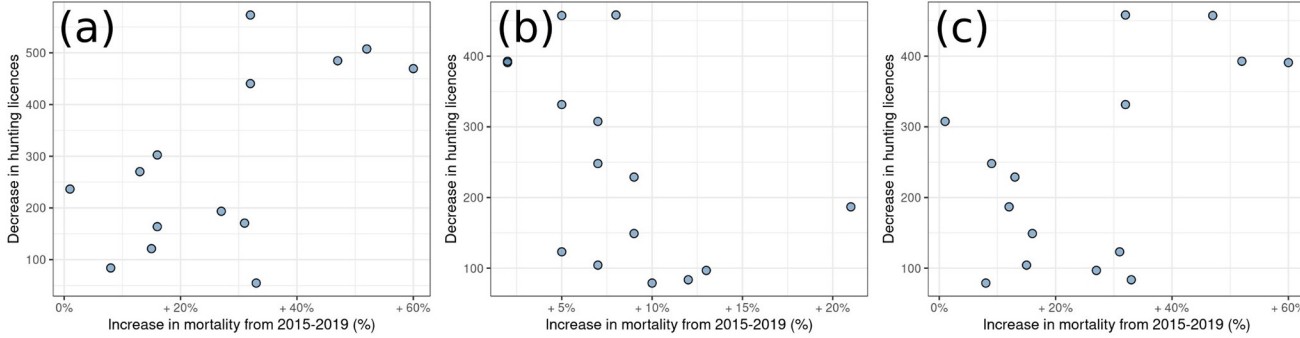

**Fig 4. Relationship between the decrease in hunting licences and excess mortality, in each province: Relationship in 2020–2021 (a), in 2021–2022 (b) and between 2021–2022 licences and 2020 excess mortality (c).** Anomalies are expressed as the difference between the median value of the posterior distribution, which in Bayesian models represent the most probable value, and observed values. To assist the interpretation of the plots the decrease in hunting licences was expressed as an absolute value. Administrative boundaries were downloaded from the Italian National Institute of Statistics and are subjected to a CCBY 4.0 license (https://www.istat.it/note-legali/).

intensive care units, particularly during spring 2020. Consequently, a higher-than-usual number of hunters may have perished from COVID-19 as they were in the most affected demographic segment.

Despite the current estimates of COVID-19 incidence are subject to debate, these areas likely experienced a higher total burden of COVID-19 in terms of incidence [45]. Hunters may have refrained from the hunting season to provide assistance to their relatives [20, 21] or because affected by long-COVID symptoms that could persists for months after the infection [54]. In turn, these two dynamics may have been mutually reinforcing, leading to the rapid dissolution of established social networks within the hunting community: as some hunters died, and others quit hunting, more hunters could have simply not renewed their licenses as their social circle was destroyed. The impact of social circle disruption, and the prohibition of gathering in groups from local health authorities, could have significantly affected forms of hunting, like wild boar drives with hounds [34], where teams must attain a minimum number of participants in order for hunting to take place. Finally, hunters might have ceased hunting due to uncertain about the opening of the hunting season, especially during 2020, as sanitary restrictions caused a confused situation where no firm decision were anticipated. Our data also suggest that these mechanisms were concentrated in time. Those provinces of the Lombardy and Emilia-Romagna regions, with the highest excess mortality in 2020 also had the highest anomalous decrease in hunting licences during the 2021/2022 season (Fig 3). This point is confirmed by considering the association between the reduction of hunting licences and excess mortality. As shown in the results, in 2020 the decrease in licences correlated positively with excess mortality, but this relationship was negative in 2021 and indeed the decrease in licences during 2021 correlated positively with 2020 excess mortality. Therefore, most impacts on hunters, their relatives and social circles, probably occurred during the first wave of COVID-19 in 2020, which made withdrawn people in 2020 and 2021. This dynamic can be considered analogous to the "harvesting effect" of pandemics, which is the compensatory reduction in mortality, which follows a temporary increase in the number of deaths among the most at-risk individuals.

The relative weight of these mechanisms is open to debate. In Italy restrictions, such as the lockdown in spring 2020, were implemented at the national level and therefore, if uncertainty had really played a major role, we should have observed an anomalous decrease across the entire study area. Future studies, carried out through structured questionnaires administered to a representative sample of hunters [55], will be crucial for delving into the ways in which COVID-19 affected the quality of the hunting experience. These investigations will help elucidate how each one of these four mechanisms could have affected the behaviour of hunters. Moreover, future studies should also replicate our analyses with long-term data, whenever these are available from regional offices: it is possible that the observed trends in 2020 and 2021 could have reversed in 2022, with some hunters renewing their hunting licenses.

However, we believe that the anomalies we observed in 2020 and 2021 should raise an alarm regarding the potential long-term consequences of COVID-19 for wildlife management in Italy. In this country, wildlife management relies almost entirely on the voluntary work of hunters, and professional hunting is forbidden by the legal framework (Legislative decree n. 157/1992). Voluntary cooperation from hunters will almost certainly collapse as they withdraw from hunting, a dynamic which COVID-19 will be further accelerated. We believe that updating existing laws to allow for the professionalization of hunting could partially counteract this collapse, by creating a corps of gamekeepers to assist local governments in wildlife management actions. Moreover, it will be necessary to reconsider management praxis to cope with a more aged and limited hunters population that could not afford physically demanding tasks.

Our findings also raise an alarm for those parts of Europe where COVID-19 had a high incidence and mortality since late 2020 [56]. In these areas, where hunters are already declining [57], a further decrease in recreational hunting could undermine the management of wild ungulates, whose populations have expanded [34] and nowadays require intensive culling [58, 59], as well as the integrated management of African Swine Fever [60]. To better understand the severity of this risk it is urgent to create a pan-European dataset about hunting licenses in Europe, to model large-scale trends in recreational hunting. This initiative will be fundamental to navigate potentially rapid changes in existing models of wildlife management and to design policies aimed at minimizing social conflicts related to wildlife.

## Acknowledgments

We express our gratitude to the people and the institutions that provided us with access to data on hunters: Alessandra Berto and the Piedmont Region "Settore Conservazione e Gestione Fauna Selvatica e Acquicoltura", CSI Piemonte, Dario Colombi (Friuli Venezia-Giulia Region), Guido Lavazza and Stefano Omizzolo of the Veneto Region and the staff of the "Territorial Services for Agriculture, Hunting and Fishing" (STACP is the Italian acronym) of the Emilia-Romagna Region.

## Author Contributions

**Conceptualization:** Jacopo Cerri, Carmela Musto, Marco Ferretti, Marco Apollonio.

**Data curation:** Jacopo Cerri, Carmela Musto, Marco Ferretti, Francesco Bisi, Clara Tattoni.

**Formal analysis:** Jacopo Cerri.

**Funding acquisition:** Mauro Delogu, Marco Apollonio.

**Investigation:** Jacopo Cerri, Carmela Musto, Marco Ferretti, Mauro Delogu, Sandro Bertolino, Adriano Martinoli, Francesco Bisi, Damiano Giovanni Preatoni, Clara Tattoni, Marco Apollonio.

**Methodology:** Jacopo Cerri, Carmela Musto, Marco Ferretti, Clara Tattoni.

**Project administration:** Jacopo Cerri.

**Resources:** Carmela Musto, Marco Ferretti, Sandro Bertolino, Adriano Martinoli, Francesco Bisi, Damiano Giovanni Preatoni.

**Software:** Jacopo Cerri.

**Supervision:** Mauro Delogu, Sandro Bertolino, Marco Apollonio.

**Validation:** Jacopo Cerri, Carmela Musto, Marco Ferretti, Mauro Delogu, Sandro Bertolino, Adriano Martinoli, Francesco Bisi, Damiano Giovanni Preatoni, Clara Tattoni, Marco Apollonio.

**Visualization:** Jacopo Cerri, Carmela Musto, Marco Ferretti, Francesco Bisi, Clara Tattoni, Marco Apollonio.

**Writing – original draft:** Jacopo Cerri, Carmela Musto, Francesco Bisi, Clara Tattoni.

**Writing – review & editing:** Jacopo Cerri, Carmela Musto, Marco Ferretti, Mauro Delogu, Sandro Bertolino, Adriano Martinoli, Francesco Bisi, Damiano Giovanni Preatoni, Clara Tattoni, Marco Apollonio.

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
