## [Decision Letter · Decision Letter 0]

28 Jun 2024

PONE-D-24-18605COVID-19 could accelerate the decline in recreational hunting: a natural experiment from Northern ItalyPLOS ONE

Dear Dr. Bisi,

Thank you for submitting your manuscript to PLOS ONE. After careful consideration, we feel that it has merit but does not fully meet PLOS ONE’s publication criteria as it currently stands. Therefore, we invite you to submit a revised version of the manuscript that addresses the points raised during the review process.

We look forward to receiving your revised manuscript.

Kind regards,

Mattias Gaglio, PhD

Academic Editor

PLOS ONE

Journal Requirements:

2. Thank you for stating the following financial disclosure: "The co-author Carmela Musto was partially supported by a research grant funded by the Vienna Science and Technology Fund (WWTF) [10.47379/ESR20009]."  

3. We note that Figures 1 and 2 in your submission contain [map/satellite] images which may be copyrighted. All PLOS content is published under the Creative Commons Attribution License (CC BY 4.0), which means that the manuscript, images, and Supporting Information files will be freely available online, and any third party is permitted to access, download, copy, distribute, and use these materials in any way, even commercially, with proper attribution. For these reasons, we cannot publish previously copyrighted maps or satellite images created using proprietary data, such as Google software (Google Maps, Street View, and Earth). For more information, see our copyright guidelines: http://journals.plos.org/plosone/s/licenses-and-copyright.

a. You may seek permission from the original copyright holder of Figures 1 and 2 to publish the content specifically under the CC BY 4.0 license.  

Reviewers' comments:

Reviewer's Responses to Questions

**Comments to the Author**

1. Is the manuscript technically sound, and do the data support the conclusions?

Reviewer #1: Partly

Reviewer #2: Yes

2. Has the statistical analysis been performed appropriately and rigorously? 

Reviewer #1: Yes

Reviewer #2: Yes

3. Have the authors made all data underlying the findings in their manuscript fully available?

Reviewer #1: Yes

Reviewer #2: Yes

4. Is the manuscript presented in an intelligible fashion and written in standard English?

Reviewer #1: Yes

Reviewer #2: Yes

5. Review Comments to the Author

Reviewer #1: This paper addresses the impacts of COVID-19 on the number of hunting licences in selected regions of Italy over approximately a decade. The text is very well written and the conclusions are straightforward. The authors demonstrate a large and immediate decrease of hunting licences upon the emergence of COVID-19. The article complies with the seven criteria used by PLoS ONE for a publication to be accepted.

Major comments:

Objective iii) has not been addressed explicitly in this paper. It should be straightforward to add one predictor to the model describing the intensity of the COVID-19 epidemic at a province level, and support the claims in lines 168-171.

The large albeit heterogeneous reduction of the number of hunting licences after onset of COVID-19 begs the question of what happened with the relative abundance of ungulate species targeted by recreational hunting in these provinces. Is there any indication that ungulate numbers went up?

Minor comments:

lines 196-198: I don't fully understand the meaning of this sentence. Typos?

Reviewer #2: This paper assesses the decline in hunting (licenses) in relation to the COVID-epidemic in Italy. The author note a decline in hunting, while it is the most important tool in ungulate monitoring and management in Italy. They indicate a concern how epidemics can reduce hunting, and call for further research on the topic in Europe (and beyond). This is an important topic to raise, as research on it has been limited. The writing, analysis and conclusions are sound and important, and therefore I would advise to accept this manuscript, with almost no additional changes needed. Underneath I mark a small amount of minor comments.

While the work shows a decline in hunting in relation to the COVID-lockdown, the statistical methods did not link the impacts of covid directly to the decline in hunting. L169-171 + 176-178: Would there be a (statistical) way to link the impacts of covid (e.g. death rate) to the observed hunter declines?

The paper is very well written and concise. I did not detect spelling or grammatical errors and the writing is very much to the point.

L175: ‘… where hunters are aging…’ : Are your results only relevant for aging populations of hunters? I don’t think this can be deducted from the data. But I agree with the suggested mechanisms.

L208-216: I agree that hunting may have picked up (hunters renewing their licenses) after the epidemic. I think the paper shows that hunting declines during an epidemic, but there is no reason to assume that this does not pick up again afterwards (unless due to death of hunters in an aging population), as many other human activities have been shown to do. However, I do agree with the increased need/role of hunters in managing ungulate populations. Perhaps indicating that hunting should be professionalized? I agree with the message for further research.

6. PLOS authors have the option to publish the peer review history of their article (what does this mean?). If published, this will include your full peer review and any attached files.

Reviewer #1: No

Reviewer #2: **Yes: **Bjorn Mols

---

## [Author Response · Author response to Decision Letter 0]

24 Jul 2024

Rebuttal letter

PONE-D-24-18605: “COVID-19 could accelerate the decline in recreational hunting: a natural experiment from Northern Italy”

Dear Editor,

please find all the responses in the following letter

################################################################################

#Editor’s comments

################################################################################

Comment: Please ensure that your manuscript meets PLOS ONE's style requirements, including those for file naming. The PLOS ONE style templates can be found at https://journals.plos.org/plosone/s/file?id=wjVg/PLOSOne_formatting_sample_main_body.pdf and 

https://journals.plos.org/plosone/s/file?id=ba62 PLOSOne_formatting_sample_title_authors_affiliations.pdf

Reply: we followed the published guidelines

Comment: Thank you for stating the following financial disclosure: "The co-author Carmela Musto was partially supported by a research grant funded by the Vienna Science and Technology Fund (WWTF) [10.47379/ESR20009]." Please state what role the funders took in the study. If the funders had no role, please state: ""The funders had no role in study design, data collection and analysis, decision to publish, or preparation of the manuscript."" If this statement is not correct you must amend it as needed. Please include this amended Role of Funder statement in your cover letter; we will change the online submission form on your behalf.

Reply: we modified the financial disclosure as you suggested and we reported the new sentence in the Cover Letter. "The co-author Carmela Musto was partially supported by a research grant funded by the Vienna Science and Technology Fund (WWTF) [10.47379/ESR20009], the funders had no role in study design, data collection and analysis, decision to publish, or preparation of the manuscript."

Comment: We note that Figures 1 and 2 in your submission contain [map/satellite] images which may be copyrighted. All PLOS content is published under the Creative Commons Attribution License (CC BY 4.0), which means that the manuscript, images, and Supporting Information files will be freely available online, and any third party is permitted to access, download, copy, distribute, and use these materials in any way, even commercially, with proper attribution. For these reasons, we cannot publish previously copyrighted maps or satellite images created using proprietary data, such as Google software (Google Maps, Street View, and Earth). For more information, see our copyright guidelines: http://journals.plos.org/plosone/s/licenses-and-copyright.

You may seek permission from the original copyright holder of Figures 1 and 2 to publish the content specifically under the CC BY 4.0 license. We recommend that you contact the original copyright holder with the Content Permission Form (http://journals.plos.org/plosone/s/file?id=7c09/content-permission-form.pdf) and the following text:“I request permission for the open-access journal PLOS ONE to publish XXX under the Creative Commons Attribution License (CCAL) CC BY 4.0 (http://creativecommons.org/licenses/by/4.0/). Please be aware that this license allows unrestricted use and distribution, even commercially, by third parties. Please reply and provide explicit written permission to publish XXX under a CC BY license and complete the attached form.” Please upload the completed Content Permission Form or other proof of granted permissions as an ""Other"" file with your submission. In the figure caption of the copyrighted figure, please include the following text: “Reprinted from [ref] under a CC BY license, with permission from [name of publisher], original copyright [original copyright year].”

Reply: we used polygons from the Italian National Institute of Statistics, which are covered by a CCBY 4.0 license. We now specified this in the caption of each figure.

Comment: Please review your reference list to ensure that it is complete and correct. If you have cited papers that have been retracted, please include the rationale for doing so in the manuscript text, or remove these references and replace them with relevant current references. Any changes to the reference list should be mentioned in the rebuttal letter that accompanies your revised manuscript. If you need to cite a retracted article, indicate the article’s retracted status in the References list and also include a citation and full reference for the retraction notice.

Reply: we reviewed the reference list and no changes have been made

################################################################################

Review Comments to the Author - Reviewer #1:

################################################################################

Comment: This paper addresses the impacts of COVID-19 on the number of hunting licences in selected regions of Italy over approximately a decade. The text is very well written and the conclusions are straightforward. The authors demonstrate a large and immediate decrease of hunting licences upon the emergence of COVID-19. The article complies with the seven criteria used by PLoS ONE for a publication to be accepted.

Reply: Thanks for appreciating our study, we addressed each one of your comments in detail.

Comment: Objective iii) has not been addressed explicitly in this paper. It should be straightforward to add one predictor to the model describing the intensity of the COVID-19 epidemic at a province level, and support the claims in lines 168-171.

Reply: we agree with your suggestion. Now we explored the association between anomalous variations in hunting licences during 2020 and 2021, and the associated excess mortality in each year. We downloaded excess mortality estimates from the Italian National Institute of Statistics (https://www.istat.it/it/files//2022/03/Report_ISS_ISTAT_2022_tab3.pdf) and plotted them on a map. Moreover, we used Spearman’s correlation to test for the association between anomalies in hunting licences and excess mortality in 2020, 2021 and between 2021 licences and 2020 mortality. This analysis tested for hypothesis iii) and also revealed the probable occurrence of the “harvesting effect”, with most impacts on hunters having occurred in 2020 and preventing further decrease in 2021. Please see lines 180 – 183 and Figure 4 and additional comments in the discussion chapter.

Comment: The large albeit heterogeneous reduction of the number of hunting licences after onset of COVID-19 begs the question of what happened with the relative abundance of ungulate species targeted by recreational hunting in these provinces. Is there any indication that ungulate numbers went up?

Reply: unfortunately, we do not possess enough information to answer this question, which would be absolutely fascinating. In the study area, wild ungulate populations are estimated with methods and sampling schemes that differ between the various regions. Therefore, even in the pre-pandemic period was already hard to compare region-level estimates. This problem was exhacerbated by the fact that in 2020, regional offices often suspended large-scale population monitoring, due to the difficulty in engaging voluntary hunters, while complying with sanitary restrictions. Also neither roadkill data, nor hunting bags are an option: in the study area there is no systematic collection of roadkill (https://doi.org/10.1111/1365-2664.14140) and the interpretation of hunting bags, across short temporal scales, relies on adjusting them for hunting effort. Which is impossible to quantify as regions do not collect effort measures from hunters. Finally, it is worth mentioning that any future interpretation of post-COVID19 trends for the wild boar might be influenced by the current epidemic of African Swine Fever, that started in January 2022 (https://doi.org/10.1111%2Ftbed.14584).

Comment: lines 196-198: I don't fully understand the meaning of this sentence. Typos?

Reply: we now rephrased this sentence, please see lines 266-269.

################################################################################

Review Comments to the Author - Reviewer #2

################################################################################

Comment: This paper assesses the decline in hunting (licenses) in relation to the COVID-epidemic in Italy. The author note a decline in hunting, while it is the most important tool in ungulate monitoring and management in Italy. They indicate a concern how epidemics can reduce hunting, and call for further research on the topic in Europe (and beyond). This is an important topic to raise, as research on it has been limited. The writing, analysis and conclusions are sound and important, and therefore I would advise to accept this manuscript, with almost no additional changes needed. Underneath I mark a small amount of minor comments.

Reply: thanks for your appreciation. We addressed each one of your comments in detail.

Comment: While the work shows a decline in hunting in relation to the COVID-lockdown, the statistical methods did not link the impacts of covid directly to the decline in hunting. L169-171 + 176-178: Would there be a (statistical) way to link the impacts of covid (e.g. death rate) to the observed hunter declines?

Reply: we now explored the association between excess deaths in 2020 and 2021 and the decrease in hunting licences among provinces. We downloaded excess mortality estimates from the Italian National Institute of Statistics (https://www.istat.it/it/files//2022/03/Report_ISS_ISTAT_2022_tab3.pdf) and plotted them on a map. Moreover, we used Spearman’s correlation to test for the association between anomalies in hunting licences and excess mortality in 2020, 2021 and between 2021 licences and 2020 mortality. This analysis tested for hypothesis iii) and also revealed the probable occurrence of the “harvesting effect”, with most impacts on hunters having occurred in 2020 and preventing further decrease in 2021. Please see lines 180 – 183 and Figure 4 and additional comments in the discussion chapter.

Comment: The paper is very well written and concise. I did not detect spelling or grammatical errors and the writing is very much to the point.

Reply: thanks.

Comment: L175: ‘… where hunters are aging…’ : Are your results only relevant for aging populations of hunters? I don’t think this can be deducted from the data. But I agree with the suggested mechanisms.

Reply: It cannot be deducted from our data, but we explained in the “Materials and methods” that most hunters in the study area are older than 60 years of age (see lines 99-101).

Comment: L208-216: I agree that hunting may have picked up (hunters renewing their licenses) after the epidemic. I think the paper shows that hunting declines during an epidemic, but there is no reason to assume that this does not pick up again afterwards (unless due to death of hunters in an aging population), as many other human activities have been shown to do. However, I do agree with the increased need/role of hunters in managing ungulate populations. Perhaps indicating that hunting should be professionalized? I agree with the message for further research.

Reply: Even though the current legal framework in Italy does not allow for professional hunting, we now included your recommendation in the manuscript. We agree with it, as hunting will almost certainly keep declining in the near future, and the current wildlife management model, that relies entirely on volunteering, is unsustainable. Alternatives are needed and the professionalization of hunting might be one of that. Please see lines 255 - 264.

Comment: 6. PLOS authors have the option to publish the peer review history of their article (what does this mean?). If published, this will include your full peer review and any attached files. Reply: Yes, we would like to have the peer review history of our article published.

Comment: While revising your submission, please upload your figure files to the Preflight Analysis and Conversion Engine (PACE) digital diagnostic tool, https://pacev2.apexcovantage.com/. PACE helps ensure that figures meet PLOS requirements. To use PACE, you must first register as a user. Registration is free. Then, login and navigate to the UPLOAD tab, where you will find detailed instructions on how to use the tool. If you encounter any issues or have any questions when using PACE, please email PLOS at figures@plos.org. Please note that Supporting Information files do not need this step.

Reply: we used PACE software to check and fix figures.

---

## [Decision Letter · Decision Letter 1]

5 Aug 2024

COVID-19 could accelerate the decline in recreational hunting: a natural experiment from Northern Italy

PONE-D-24-18605R1

Dear Dr. Bisi,

We’re pleased to inform you that your manuscript has been judged scientifically suitable for publication and will be formally accepted for publication once it meets all outstanding technical requirements.

Kind regards,

Mattias Gaglio, PhD

Academic Editor

PLOS ONE

Additional Editor Comments (optional):

Reviewers' comments:

Reviewer's Responses to Questions

**Comments to the Author**

1. If the authors have adequately addressed your comments raised in a previous round of review and you feel that this manuscript is now acceptable for publication, you may indicate that here to bypass the “Comments to the Author” section, enter your conflict of interest statement in the “Confidential to Editor” section, and submit your "Accept" recommendation.

Reviewer #1: All comments have been addressed

2. Is the manuscript technically sound, and do the data support the conclusions?

Reviewer #1: Yes

3. Has the statistical analysis been performed appropriately and rigorously? 

Reviewer #1: (No Response)

4. Have the authors made all data underlying the findings in their manuscript fully available?

Reviewer #1: (No Response)

5. Is the manuscript presented in an intelligible fashion and written in standard English?

Reviewer #1: (No Response)

6. Review Comments to the Author

Reviewer #1: All revisions requested have been addressed properly, and the paper is in good shape for publication.

7. PLOS authors have the option to publish the peer review history of their article (what does this mean?). If published, this will include your full peer review and any attached files.

Reviewer #1: No

---

## [Editor Report · Acceptance letter]

8 Aug 2024

PONE-D-24-18605R1 

PLOS ONE

Dear Dr. Bisi, 

I'm pleased to inform you that your manuscript has been deemed suitable for publication in PLOS ONE. Congratulations! Your manuscript is now being handed over to our production team.

Kind regards, 

on behalf of

Dr. Mattias Gaglio 

Academic Editor

PLOS ONE